# A Multi-Faceted Approach to Tuberculosis Active Case Finding among Remote Riverine Communities in Southern Nigeria

**DOI:** 10.3390/ijerph18189424

**Published:** 2021-09-07

**Authors:** Andy Samuel Eyo, Valerie Okon Obot, Okezie Onyedinachi, Nathaly Aguilera Vasquez, Jacob Bigio, Ataulhaq Sanaie, Favour Beulah, Uduak Ette, Dennis Uju, Md. Toufiq Rahman

**Affiliations:** 1Excellence Community Education Welfare Scheme (ECEWS), Uyo 520231, Nigeria; andy@ecews.org (A.S.E.); okezie@ecews.org (O.O.); favour@ecews.org (F.B.); uduak@ecews.org (U.E.); uju@ecews.org (D.U.); 2McGill International TB Center, Research Institute of the McGill University Health Center, Montreal, QC H3G 1A4, Canada; nathaly.aguileravasquez@mail.mcgill.ca (N.A.V.); jacob.bigio@affiliate.mcgill.ca (J.B.); 3TB REACH External M&E Reviewer, London E11 4DP, UK; Sanaie.as@gmail.com; 4Stop TB Partnership, 1218 Geneva, Switzerland; toufiqr@stoptb.org

**Keywords:** tuberculosis, active case finding, community-based

## Abstract

Nigeria accounts for 11% of the worldwide gap between estimated and reported individuals with tuberculosis (TB). Hard-to-reach communities on the Southern Nigeria coast experience many difficulties accessing TB services. We implemented an active case finding (ACF) intervention in Akwa Ibom and Cross River states utilizing three approaches: house-to-house/tent-to-tent screening, community outreach and contact investigation. To evaluate the impact, we compared TB notifications in intervention areas to baseline and control population notifications, as well as to expected notifications based on historical trends. We also gathered field notes from discussions with community volunteers who provided insights on their perspectives of the intervention. A total of 509,768 individuals were screened of which 12,247 (2.4%) had TB symptoms and 11,824 (96.5%) were tested. In total, 1015 (8.6%) of those identified as presumptive had confirmed TB—98.2% initiated treatment. Following implementation, TB notifications in intervention areas increased by 112.9% compared to baseline and increased by 138.3% when compared to expected notifications based on historical trends. In contrast, control population notifications increased by 101% and 49.1%, respectively. Community volunteers indicated a preference for community outreach activities. Multi-faceted, community-based interventions in Nigeria’s coastal areas successfully increase TB detection for communities with poor access to health services.

## 1. Introduction

Despite global declines in tuberculosis (TB) incidence and deaths, an estimated 10.0 million people fell ill with TB and over 1.2 million perished from the disease in 2019 [1]. The End TB strategy targets a 90% reduction in TB mortality and an 80% reduction in TB incidence by 2030 compared to 2015 levels [2]. One of the bottlenecks in achieving these targets stems from individuals falling through the cracks of official case reporting systems set up by National TB Programs—often referred to as the “missing millions”—who consequently cannot access quality TB care [3]. There are many reasons for this, including weak health systems, poor knowledge and awareness on TB, poor linkages to the private sector and lack of community engagement [3].

Nigeria has one of the world’s largest gaps between estimated and reported individuals with TB, contributing to 11% of the world’s missing millions, with only 117,300 (26.7%) out of an estimated 440,000 newly diagnosed individuals reported to the National Tuberculosis and Leprosy Control Program (NTBLCP) in 2019 [1]. The NTBLCP largely relies on passive case finding (PCF) to identify individuals with TB, an approach which only identifies people with TB who present with symptoms at health facilities [4]. However, active case finding (ACF) initiatives such as house-to-house screening are important to reach undiagnosed individuals not presenting to health facilities [5]. This approach utilizes a provider-initiated approach where at-risk populations are targeted through various systematic screening approaches.

In the coastal areas of Southern Nigeria, the population is mainly comprised of fishing and farming communities and settlements which are often subject to high levels of poverty [6]. These communities are considered hard-to-reach due to poor road infrastructure and being surrounded by creeks requiring boats to access certain locations [7,8]. As a result, riverine communities face many barriers in accessing health care such as unavailability of or difficulty reaching health facilities, lack of funds to finance health services or health facilities that lack appropriate medication or trained staff [8]. Further, poverty, illiteracy, as well as cultural and religious beliefs in these communities often impede health seeking [7]. It has been previously noted that hard-to-reach populations often have to walk over an hour to reach a health facility, and studies have shown that women living in the Atlantic coastline of Nigeria have poor awareness of TB [9,10].

In order to improve access to health services and TB case detection, Excellence Community Education Welfare Scheme (ECEWS), a non-profit organization aiming to improve health and education in Nigeria, received a proof-of-concept grant from the TB REACH Initiative of the Stop TB Partnership to implement a multi-faceted ACF intervention in riverine and hard-to-reach communities in two states on the Southern Nigerian coastline.

## 2. Methods

### 2.1. Intervention

The intervention was implemented in the lbeno, Oron and Mbo local government areas (LGAs) of Akwa Ibom State and in the Biase, Obubra and Odukpani LGAs of Cross River State (henceforth evaluation population) (Figure 1). The evaluation population was selected based on its location in Nigeria’s coastal areas. Three control LGAs were purposively selected based on similarity of the population to the evaluation population to allow for a pre-post comparison with a comparable population to that of evaluation population, Control LGAs were designated in Rivers State, namely Etche, Oyigbo, Ikwerre and Tai (henceforth control population) (Figure 1). The selection of the control population was based on similarity to the evaluation population in (1) population size and (2) access to health services (based on number of health facilities in the area). Further, control population was selected in a different state (Rivers State) to limit individuals from filtering into the evaluation population.

Three distinct approaches were employed to increase the reach of the intervention: house-to-house/tent-to-tent (H2H/T2T) screening, targeted community outreach and household contact investigation. ECEWS engaged community volunteers (CVs), patent medicine vendors (PMVs) and community pharmacists (CPs) to carry out activities. All were trained in general TB knowledge and screening, as well as sputum collection. In the H2H/T2T intervention, CVs verbally screened members of consenting households. For targeted community outreach, CVs screened participants in town halls, village squares, markets, local car parks and churches, while PMVs and CPs screened clients in their shops. Household contact investigation was carried out by CVs upon obtaining contact information from individuals diagnosed with TB. To raise awareness on the intervention, the project disseminated information through print media and provided health talks in schools. To encourage screening of women and children, healthcare workers (HCWs) also held weekly TB awareness workshops and verbal screening for pregnant women and children in antenatal clinics (ANCs).

Individuals with presumptive TB were defined as individuals who presented TB-like symptoms, namely ≥2 weeks of cough, night sweats, weight loss and/or fever. Upon identifying individuals with TB-like symptoms, CVs, PMVs or CPs obtained sputum samples and communicated with riders who transported samples using medical cooling boxes to general hospitals or stand-alone laboratories within the evaluation LGAs. All sputum samples were tested using GeneXpert MTB/RIF on same day of collection or within 48 h. In order to facilitate Xpert testing, ECEWS installed a GeneXpert MTB/RIF machine in Obubra Hospital. In all other LGAs, GeneXpert machines were available in close proximity to intervention communities. On occasions when CVs had access to motorbikes, they also took on the role of riders. If an individual identified as presumptive was not able to produce a sputum sample on the spot, they were provided with a specimen tube labelled with their name that a CV could collect the next morning from their homes. To diagnose childhood TB, the project engaged nearby health facilities and covered the cost of the chest X-ray (CXR) and transport for the parent and child. Childhood TB diagnosis was determined by the on-site clinician whose services were also compensated by ECEWS to ensure timely evaluation of CXR. To ensure safety of all project staff, all involved in screening and collection of sputum samples were provided face masks, gloves, and antiseptic solution.

Results were retrieved by the riders or CVs and shared with the LGA TB and Leprosy Supervisors (LGTBLS) who subsequently linked individuals with confirmed TB to Directly Observed Treatment Short Course (DOTS) centers. Throughout the intervention, CVs, PMVs and CPs provided treatment support for individuals who requested it or lived far from DOTS centers by delivering the medication through home visits (CVs) or providing it in their shops (PMVs, CPs)—with the approval of the LGTBLS. The CVs, PMVs and CPs were given performance-based incentives for each individual with bacteriologically confirmed (Bac+) TB identified.

To encourage acceptance of the intervention, ECEWS engaged four community leaders in each intervention LGA. Engaged community leaders were selected among community gatekeepers (e.g., village heads, women and youth leaders, pastors) based on willingness to participate. Community leaders served as local representatives of the ECEWS intervention by coordinating screening activities and advocating for the intervention in the engaged communities.

### 2.2. Data Collection

During verbal screening, CVs, PMVs and CPs were provided paper-based TB symptom checklists and referral registers. All sputum samples collected were recorded using a sputum examination and request form. Individuals with presumptive TB were documented in the health facility’s presumptive TB register and information on sputum samples was entered into laboratory’s registers which also captured the result of each test. All individuals with confirmed TB were notified to the LGTBLS and were also documented in DOTS center treatment registers for treatment initiation. ECEWS Monitoring and Evaluation officers gathered and validated all data. TB notification data for the intervention period, as well as historical data was collected for both the evaluation LGAs and control LGAs from State TB program notification systems. All aggregate data were collected and tabulated on Excel 2016. Field notes were also gathered from discussions with CVs to gain their insights into the intervention.

### 2.3. Data Analysis

Quantitative data were analyzed according to the TB REACH framework as previously reported [11]. The baseline period was determined by notification data for the period of 1 October 2017 to 30 September 2018. Since the ECEWS intervention was conducted between 1 October 2018 to 31 December 2019 which incorporates an additional fifth quarter, baseline data (collected over four quarters) were multiplied by a factor of 1.25 in order to account for the additional quarter. Notification data from the evaluation population were compared with baseline and control population notification data. A linear regression using 3-year historical notification data from 1 October 2015 to 30 September 2018 were used to extrapolate the expected case notifications for the evaluation and control populations for the intervention period. This enabled a comparison between case notifications during the implementation period and the expected notifications if the intervention had not been implemented. Baseline data enabled computation of unadjusted additional notifications which is calculated by: (total notifications after implementation)—(total notifications during baseline period). To consider historical trends in notifications, adjusted additional notifications was computed through considering the number of notifications that would have occurred without the intervention (i.e., expected notifications), through the following: (total notifications after implementation)—(3-year trend-adjusted expected notifications). Expected notifications were computed using the “FORECAST” function on Excel 2016 which predicts values based on a trend line.

## 3. Results

### 3.1. Intervention Notification Data

ECEWS trained and engaged a total of 120 CVs (70 female, 50 male), 111 PMVs (30 female, 81 male), 9 CPs (4 female, 5 male), 30 HCWs (all female) and 24 community leaders (12 female, 12 male). Between 1 October 2018 to 31 December 2019, a total of 509,768 individuals were screened (Table 1). Of those screened, 12,247 (2.4%) had one or more TB symptoms and 11,824 (96.5%) were tested. In total, 1015 (8.6%) of those identified as presumptive had confirmed TB—all forms including clinically confirmed, Bac+ and extra-pulmonary TB—of which 808 (79.6%) had Bac+ TB. The majority (98.2%) of individuals with confirmed TB initiated treatment, and all those who initiated treatment also completed treatment. Table 1 further outlines results of the intervention by gender. Overall, more women were screened for TB than men, but a larger proportion of men had confirmed TB (Bac+ and all forms) with a male to female ratio of 1.1.

Table 2 compares total TB case notifications for the control and evaluation population during the implementation period with notifications during the baseline period of 1 October 2017 to 30 September 2018, as well as with expected notifications for the implementation period. For all forms TB, notifications increased by 112.9% compared baseline notification data and by 138.3% compared to expected notifications during the implementation period in the evaluation population while in the control population notifications increased by 101.0% and 49.1%, respectively. This is further reflected in Figure 2 which showcases an upward trend in notifications in the evaluation population (y = 8.826x + 85.096) which is stronger than that of the control population (y = 2.3162x + 41.566). Thus, although historical notifications were higher in the evaluation population and both control and evaluation population had notifications higher than the expected notifications, the difference was substantially higher in the evaluation population. Table 3 outlines the additional childhood TB notifications compared to baseline for the evaluation population which increased by 656% following the intervention.

Table 4 outlines the results of the intervention by each approach employed by ECEWS. Most were screened either through community outreach or H2H/T2T. Contact screening resulted in a higher proportion of individuals identified as presumptive (11.5% compared to 2.2% for the community outreach and H2H/T2T). For other indicators, yield was similar throughout all three approaches. However, the number needed to screen (NNS) for contact investigation was substantially lower than for community outreach and H2H/T2T.

### 3.2. Observations from the Field

#### 3.2.1. Impact and Reception of the Intervention

Overall, CVs indicated a positive experience working for the ECEWS intervention. CVs shared that training helped increase their knowledge on TB which enabled them to forego traditional beliefs such as TB being caused by witchcraft. Many indicated that benefits of the intervention were that they could support their communities and be recognized for their work. They also noted the importance of living within the communities they were working with and how this enabled higher acceptance of screening. CVs shared that the intervention not only helped increase awareness of TB in the community, but it also shifted negative perspectives of TB (i.e., TB seen as a punishment or always resulting in death). However, CVs reported resistance towards sputum collection and persistence of stigma towards TB in the community. It was also noted that community leaders were very supportive and encouraging of the intervention and even directly referred community members to CVs for screening.

#### 3.2.2. Perception of the Three Different Approaches

Many CVs indicated that the H2H/T2T intervention enabled them to be perceived as focal points for TB since household members became familiar with them. They also indicated that this approach was most beneficial for people who could not leave their homes or were bedridden. However, in H2H/T2T, CVs encountered individuals who had negative reactions to screening or sputum collection, sometimes demanding incentives such as food or beverages in return for screening. Further, some individuals feared that if they let the CV into their homes, they would be stigmatized by their neighbors who might think they had TB. Further, community outreach activities resulted in large volumes of people being screened in one event but were most successful when support was garnered from the community. Certain community members attended outreach activities but desired compensation (i.e., money, food) or wanted to receive health services for other conditions. Thus, clear communication of the objective of these activities became important during community outreach to manage expectations. For contact investigation, CVs indicated that contacts were more accepting and appreciative of the intervention because they already knew from their relative or friend with TB the benefits of receiving TB treatment. However, in instances where the relative or friend was experiencing negative treatment side effects, they could discourage their contacts from participating. Further, individuals diagnosed with TB did not always provide accurate addresses for contacts. CVs indicated a perception of community outreach as a simpler approach as it did not involve as much transport and allowed them to work with other CVs as a team.

## 4. Discussion

Through a multi-faceted intervention incorporating community outreach, H2H/T2T screening and contact investigation, ECEWS successfully increased case detection of TB in Akwa Ibom and Cross River states. The results of this intervention contribute to the existing body of evidence indicating the importance of ACF in low resource settings [4,12,13] as evidenced by the 112.9% increase in TB notifications during the implementation period compared to baseline and the 138.3% increase compared to expected notifications. The three approaches resulted in similar yields; however, anecdotal information gathered from community volunteers provided insights into the unique challenges of each approach. For instance, while H2H/T2T screening was considered useful to gain access to individuals who could not otherwise access services and it was indicated that screening through contact investigation was generally more accepted, community outreach was preferred among the CVs due it being an easier and more team-oriented approach.

Findings that more men were diagnosed with TB are in line with existing literature indicating that men are generally overrepresented in TB notification data [14]. The results of this intervention further highlight the importance of community-based approaches in increasing childhood TB notifications in Nigeria, as has been previously noted [15]. Low case notification in children stem from challenges in diagnosing childhood TB due to lack of accurate diagnostic tests for this population, often resulting in false negative results [16]. Our approach of providing transport vouchers to parents and engaging clinicians to provide clinical evaluation and CXR interpretation helped increase childhood TB notification by 656% in the evaluation population. However, the persistent low proportion of childhood TB cases relative to overall TB cases found underlines the fact that further efforts need to be made to increase access to TB care for children, as well as improve diagnosis and management of childhood TB. Further, the contact investigation approach yielded a lower NNS than the other two approaches which is consistent with evidence indicating that contacts of people with TB are at higher risk for developing TB [17]. In this project, CVs facilitated sputum collection and transport to labs which helped decrease pre-diagnostic loss to follow-up. A previous study by Vyas et al., also highlighted that community-based sputum collection and transport allows for more individuals to undergo testing than solely referring individuals with TB symptoms to laboratories [12]. In this intervention, treatment completion was extremely successful with a 100% completion rate among those who initiated treatment. One of the reasons for the success of treatment completion could be due to engagement of CVs, PMVs and CPs to provide treatment support to individuals who would otherwise have had difficulty reaching DOTS centers.

Previous literature has highlighted the importance of employing community-based approaches in Nigeria [18]. This intervention demonstrates that an emphasis should also be placed on the role of CVs who are rooted within the communities they are serving. In this intervention, engaging highly motivated CVs helped overcome many of the challenges encountered during screening such as reluctance towards screening and overcoming negative behavior in the community. Although CVs were provided with monetary incentives for their work, they indicated that supporting their community and being recognized for their work were also important factors. This is similar to the findings of Khan et al. who suggest that financial gain is not always a principal motivator for community health workers, and that intrinsic motivation stemming from moral or religious factors (i.e., desire to help others) may play an important role [19].

Another important aspect of this intervention was that strong engagement of community leaders facilitated implementation of the intervention. Involvement of local leaders to overcome resistance in the community has been noted as an important consideration for ACF interventions [20] and has been indicated as an important step in creating effective health interventions in Nigeria [7].

Throughout implementation, ECEWS accumulated many lessons learnt in regard to implementing Xpert testing in a hard-to-reach and underserved area. Although there was high Xpert coverage in the EP, the increase in volume of testing samples due to the intervention overwhelmed nearby laboratories (cartridge shortages, overworked staff). This resulted in longer turnaround times for results. To mitigate this challenge, the project staff engaged Xpert laboratories outside of intervention LGAs, which compensated for challenges in nearby laboratories. Cattamanchi et al. indicate that high Xpert coverage must be accompanied by addressing larger health system barriers, such as improved staff training and coordination, in order to ensure maximum impact on diagnostics [21]. Further, the project ensured easy access to Xpert testing in all EP and procured a GeneXpert MTB/RIF machine for the Obubra LGA where it was not previously accessible.

One of the limitations of this intervention is that although the project provided free CXR for children, the budget did not allow financing of CXR for adults, which could have increased the number of clinically diagnosed individuals. Further, the evaluation of this intervention is based on programmatic data and not on data collected in a controlled study setting, thus the increase in TB notifications during the intervention period cannot be fully attributed to the intervention. However, the use of a control population as well as baseline data does suggest improvements due to the intervention.

## 5. Conclusions

The riverine and hard-to-reach populations of Southern Nigeria face many challenges to accessing care. However, community-based interventions such as this one that support case finding in a variety of ways in the target communities can result in important gains in detecting TB. Working with local leaders and staff to ensure success of interventions is vital in this context. It is important to note that this intervention not only sought to improve case finding, but also worked to strengthen many aspects of the TB care cascade by providing various opportunities for screening, facilitating diagnosis through on-site sputum collection and transport, as well as providing sensitive diagnosis through Xpert testing, easy linkage to treatment and treatment support where required. All this was accompanied by successful engagement of hard-to-reach communities, careful supervision of the data collection process to ensure quality data, as well as consistently troubleshooting implementation challenges when they occurred. Thus, this intervention not only resulted in increased diagnosis of TB, but also in improved quality of care for all individuals engaged by the project. Due to the success of this intervention, ECEWS received an additional TB REACH grant to scale up their intervention to 15 LGAs across five states, including Akwa Ibom, Bayelsa, Cross River, Delta and Imo. Further, the ECEWS ACF model is being considered for inclusion into Nigeria’s National Strategic Plan for Tuberculosis. Similar community-based, multi-faceted approaches could be effective to reach populations with limited access to TB screening and diagnostic services in other high TB burden countries.

## Figures and Tables

**Figure 1 ijerph-18-09424-f001:**
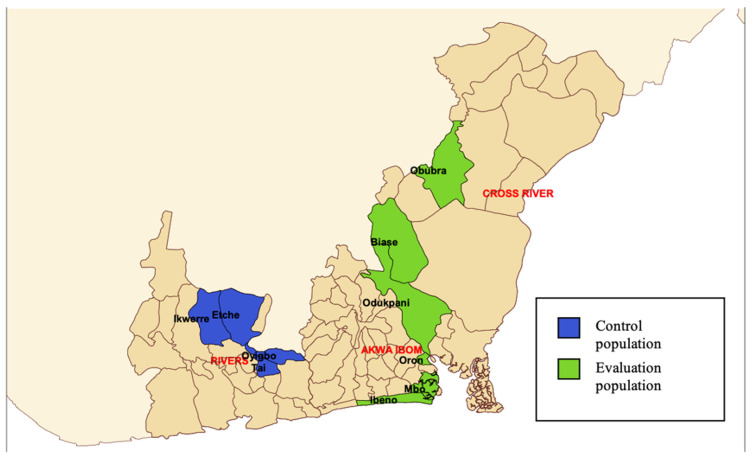
Map of selected LGAs in the control (Rivers State) and evaluation populations (Akwa Ibom and Cross River States) for the ECEWS intervention.

**Figure 2 ijerph-18-09424-f002:**
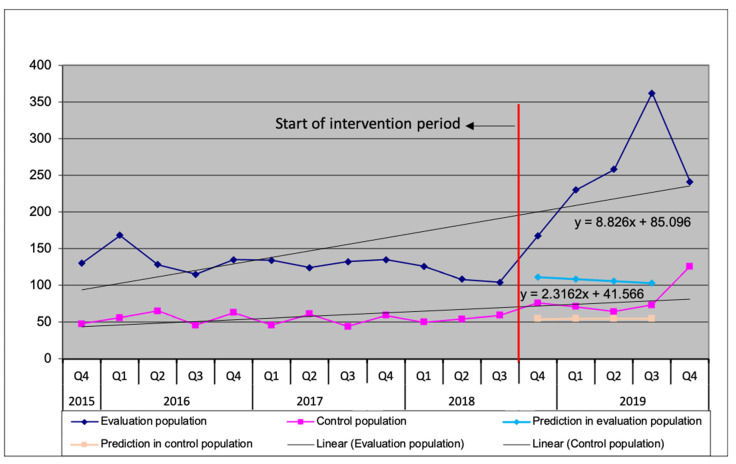
All forms TB cases during baseline and implementation periods and expected values during implementation period based on historical 3-year linear trend.

**Table 1 ijerph-18-09424-t001:** Results of the intervention in lbeno, Oron, Mbo (Akwa Ibom) and in Biase, Obubra and Odukpani (Cross River).

	Total N (%)	Male N (%)	Female N (%)
Number of people screened	509,768	241,745	268,023
Number of individuals with presumptive TB (% among screened)	12,247 (2.4)	5443 (2.3)	6804 (2.5)
Number tested for TB (% among individuals with presumptive TB)	11,824 (96.5)	5236 (96.2)	6588 (96.8)
Number of people with Bac+ ^1^ TB (% among tested)	808 (6.8)	436 (8.3)	372 (5.6)
Number of individuals diagnosed with all forms ^2^ TB (% among individuals with presumptive TB)	1015 (8.3)	539 (9.9)	476 (7.0)
Number of individuals with Bac+ TB started on treatment (% among Bac+ ^1^ TB)	794 (98.2)	432 (99.1)	362 (97.3)
Number of individuals with all forms TB started on treatment (% all forms TB)	1001 (98.6)	532 (98.7)	469 (98.5)
Number of individuals with Bac+ TB who completed treatment (% started treatment)	794 (100.0)	432 (100.0)	362 (100.0)
Number of individuals with all forms TB who completed treatment (% started treatment)	1001 (100.0)	532 (100.0)	469 (100.0)

^1^ Bac+ = bacteriologically confirmed; ^2^ all forms = bacteriologically and clinically confirmed and extra-pulmonary.

**Table 2 ijerph-18-09424-t002:** Additional TB case notifications compared to baseline and expected notifications in the evaluation and control populations.

	Evaluation Population		Control Population	
	Bac+	All Forms	Bac+	All Forms
Baseline notification data (2017–2018)	518	591	226	204
Notifications during implementation	921	1258	365	410
Unadjusted additional notifications	403	667	139	206
% change from baseline	77.8%	112.9%	61.5%	101.0%
Expected notifications (without intervention) *	534	528	225	275
Adjusted additional notifications	387	730	140	135
% change from expected notifications	72.5%	138.3%	62.2%	49.1%

* Expected notifications were extrapolated using an adjusted trend line with notification data from 1 October 2015 to 30 September 2018.

**Table 3 ijerph-18-09424-t003:** Additional childhood TB case notifications compared to baseline by age category.

	0–4 Years	5–14 Years	Total
Baseline childhood TB notifications (2017–2018)	5	11	16
Notifications during implementation	27	94	121
Additional notifications	22	83	105
% change from baseline	440%	755%	656%

**Table 4 ijerph-18-09424-t004:** Results of the intervention by approach in lbeno, Oron, Mbo (Akwa Ibom) and in Biase, Obubra and Odukpani (Cross River) by approach.

	Community Outreach	H2H/T2T ^1^ Screening	Contact Investigation
Number of people screened	209,177	269,069	11,700
Number of individuals with presumptive TB (% among screened)	4536 (2.2)	5786 (2.2)	1350 (11.5)
Number tested for TB (% among individuals with presumptive TB)	4405 (97.1)	5509 (95.2)	1350 (100.0)
Number of people with Bac+ TB (% among tested)	283 (6.4)	386 (7.0)	95 (7.0)
Number of people diagnosed with all forms TB (% among individuals with presumptive TB)	385 (8.5)	461 (8.0)	105 (7.8)
Number of individuals with Bac+ TB started on treatment (% among Bac+ TB)	278 (98.2)	377 (97.7)	95 (100.0)
Number of individuals with all forms TB started on treatment (% all forms TB)	385 (100.0)	452 (98.0)	105 (100.0)
Number of individuals with Bac+ TB who completed treatment (% started treatment)	278 (100.0)	377 (100.0)	95 (100.0)
Number of individuals with all forms TB who completed treatment (% started treatment)	385 (100.0)	452 (100.0)	105 (100.0)
Number needed to screen to diagnose one TB case (all forms)	543	583	111

^1^ H2H/T2T = house-to-house/tent-to-tent.

## Data Availability

Not applicable. Data are stored by the NTBLCP and are subject to the organization’s sharing and privacy policies.

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
