# Peer review of "A Multi-Faceted Approach to Tuberculosis Active Case Finding among Remote Riverine Communities in Southern Nigeria"

_ijerph, 2021, doi:10.3390/ijerph18189424_

Round 1

Reviewer 1 Report

Reviewer comments - ijerph-1314391
The authors have reported a study where they wanted to bring about a community level change in the way TB cases are reported, traced and diagnosed. This study seems largely in the public health domain and brings out hitherto unknown human tendencies and perceptions about TB in resource scare coastal communities of southern Nigeria. The authors have relied heavily on an outreach program on multiple levels to eventually get TB patients somehow to report and get treatment initiated. With minor corrections, I would recommend this manuscript suitable for publication.

Comments –

1) Line 34 – One of the bottlenecks “in” achieving, not “to” achieving. There are minor instances of English errors, colloquial language used which can be remedied by a round of proof reading.
2) Line 40 – Elaborate more on this 8% gap for a science audience not very familiar with TB epidemiology in Nigeria. Is it the % of cases reported? Is it the % cases unreported? I am not very sure on this myself.
3) Line 106 – How was it ensured that the multiple people like CV, PMVs, riders etc were also free of TB? Were they tested and confirmed as TB free before and after the study? Did any of them contract TB during the study? Were they given any protective gear when they did door to door/tent to tent outreach or collected sputum samples?
4) Figure 1 – Rectify legends. I cannot read the full legends.
5) Since most of the data collection was manual and by human intervention, on was such a huge initial dataset verified? Were these data double checked for accuracy and veracity? Is this data stored in any publicly available database? If so, provide a link to the original data for other researchers to access. Or if it is the policy to not release this data, please state so.
6) Table 3, line 174 - When numbers get to 700 % they start sounding ridiculous. Can you switch to a multiple method where if the notifications went from 40 from a baseline 20, you call it 2X rather than 100 % increase?
7) Line 182 – Make sure in the final version, tables are not broken across 2 separate pages.
8) Line 241 - “et al” should be italicized.
9) Line 279 – “cha-llenges” – avoid word breaking across lines.
10) One of the problems associated with TB, its reoccurrence and developing drug resistance is that patients don’t complete the full course of medications. The final outcome of this study was that a large % of people who reported and were diagnosed with TB, did initiate treatment. Is there any follow-up on how many of them successfully completed the short course treatment?

Author Response

Dear Editor,

We would like to thank you for the opportunity of publishing our manuscript in the Tuberculosis Program Implementation Special Issue in IJERPH. We are especially thankful of both reviewers who have evaluated our manuscript and provided helpful and constructive feedback that has helped improve our work. Point-to-point responses to each reviewer comment can be found below in blue.

REVIEWER 1
The authors have reported a study where they wanted to bring about a community level change in the way TB cases are reported, traced and diagnosed. This study seems largely in the public health domain and brings out hitherto unknown human tendencies and perceptions about TB in resource scare coastal communities of southern Nigeria. The authors have relied heavily on an outreach program on multiple levels to eventually get TB patients somehow to report and get treatment initiated. With minor corrections, I would recommend this manuscript suitable for publication.
Comments –

  • Line 34 – One of the bottlenecks “in” achieving, not “to” achieving. There are minor instances of English errors, colloquial language used which can be remedied by a round of proof reading. Thank you for this comment. We have proof-read the manuscript and made corrections where required.
  • Line 40 – Elaborate more on this 8% gap for a science audience not very familiar with TB epidemiology in Nigeria. Is it the % of cases reported? Is it the % cases unreported? I am not very sure on this myself. Thank you for this point. This represents Nigeria’s contribution to the “missing millions” across the globe; thus, Nigeria holds 11% [statistic has been updated according to the most recent TB report] of the individuals not reported to TB programs across the globe (only rivaled by India which holds 17% of the world’s missing millions). This has been updated in the text (line 41, page 1).
  • Line 106 – How was it ensured that the multiple people like CV, PMVs, riders etc were also free of TB? Were they tested and confirmed as TB free before and after the study? Did any of them contract TB during the study? Were they given any protective gear when they did door to door/tent to tent outreach or collected sputum samples? Project staff was not tested as they were not symptomatic, but if they developed symptoms, they would have been tested. However, no one reported symptoms throughout the duration of the intervention. The staff involved in screening and collection of sputum samples were given face masks, gloves, and antiseptic solutions. A sentence to this effect has been added on line 129 to 131, page 3.
  • Figure 1 – Rectify legends. I cannot read the full legends. This has been rectified.
  • Since most of the data collection was manual and by human intervention, on was such a huge initial dataset verified? Were these data double checked for accuracy and veracity? Is this data stored in any publicly available database? If so, provide a link to the original data for other researchers to access. Or if it is the policy to not release this data, please state so. The data is reported to Nigeria’s National TB program and preserved in their database. Data were gathered on the field, but underwent a 2-step validation system where it was first validated by ECEWS monitoring and evaluation officers and then during the TB REACH reporting process by a TB REACH monitoring and evaluation consultant. A statement has been added to the data availability section at the end of the document: “Data is stored by the NTBLCP and is subject to the organization’s sharing and privacy policies.” (lines 441-442, page 10).
  • Table 3, line 174 - When numbers get to 700 % they start sounding ridiculous. Can you switch to a multiple method where if the notifications went from 40 from a baseline 20, you call it 2X rather than 100 % increase? Thank you for the suggestion. However, we feel that given that our manuscript is following the TB REACH reporting framework, it is best to maintain % increase as this is done throughout all TB REACH projects.
  • Line 182 – Make sure in the final version, tables are not broken across 2 separate pages. Table 4 has been moved down to avoid disruption.
  • Line 241 - “et al” should be italicized. This has been done.
  • Line 279 – “cha-llenges” – avoid word breaking across lines. Thank you for noting this. This is due to the journal template format. In some instances, it was possible to correct, but for the most part it is required to maintain the word breaks.
  • One of the problems associated with TB, its reoccurrence and developing drug resistance is that patients don’t complete the full course of medications. The final outcome of this study was that a large % of people who reported and were diagnosed with TB, did initiate treatment. Is there any follow-up on how many of them successfully completed the short course treatment? Thank you for bringing this up. In this intervention, treatment completion was extremely successful with all those who initiated treatment also having completed treatment. We have added this data to Table 1 (page 5) and Table 4 (page 7).

Thank you for your consideration of our manuscript.

Best,

Dr. Valerie Obot.

Reviewer 2 Report

The manuscript ijerph-1314391 from the field of public health presents part of the results of a tuberculosis screening project in Nigeria, where active case-finding strategies are engaged to optimize the TBC detection rate. Three different approaches (1-house to house, 2-community outreach, and 3-contact investigation) were applied and compared to each other, using as a control both historical data and data from geographically and socially similar districts of Nigeria not included in the intervention.

The topic has a high reading interest, fulfills the aims and scopes of the submitted journal, provides proper ethical permission, uses clear language, logical structure, and relevant citations. 

The reviewer would like to raise some improving points:      

C001: P1L1-3: Title. Would you please rephrase the title to include the keyword "Tuberculosis" in the context of Active Case Finding

C002: P3L109: "…selected among community…" duplicate

C003: Table 3: How do you explain the low incidence of TB in children (with and w/o intervention). Were there any factors hampering the accessibility of this population fraction?

C004: P4L140-141. Data Analysis. Would you mind elaborating on the data extrapolation algorithm to predict registered TB cases based on historical data? The addition of a graph (scatter plot with regression line) would increase the readability of the manuscript. 

C005: Data acquisition / Project design. It is strongly recommended to provide some information about the randomization of (a) districts that received and did not receive the intervention (2) population groups that received different intervention strategies.

C006: The authors provide a thoughtful discussion about implementing the active screening strategies, weighing both the benefits and restraints, especially regarding the availability of screening resources. A 2-fold increase in incidence by the least intensive approach, i.e., the community outreach, is an impressive number. It would be fascinating for the readers to discuss the direct implication of this finding in the Public Health organization of Nigeria and provide a financial outlook on how the state of Nigeria or the sponsors of this research are planning to proceed.   

Author Response

Dear Editor,

We would like to thank you for the opportunity of publishing our manuscript in the Tuberculosis Program Implementation Special Issue in IJERPH. We are especially thankful of both reviewers who have evaluated our manuscript and provided helpful and constructive feedback that has helped improve our work. Point-to-point responses to each reviewer comment can be found below in blue.

REVIEWER 2

The manuscript ijerph-1314391 from the field of public health presents part of the results of a tuberculosis screening project in Nigeria, where active case-finding strategies are engaged to optimize the TBC detection rate. Three different approaches (1-house to house, 2-community outreach, and 3-contact investigation) were applied and compared to each other, using as a control both historical data and data from geographically and socially similar districts of Nigeria not included in the intervention. The topic has a high reading interest, fulfills the aims and scopes of the submitted journal, provides proper ethical permission, uses clear language, logical structure, and relevant citations. 

The reviewer would like to raise some improving points:      

  • C001: P1L1-3: Title. Would you please rephrase the title to include the keyword "Tuberculosis" in the context of Active Case Finding Thank you for noting this, “tuberculosis” has been added to the title.
  • C002: P3L109: "…selected among community…" duplicate. The duplicate has been removed.
  • C003: Table 3: How do you explain the low incidence of TB in children (with and w/o intervention). Were there any factors hampering the accessibility of this population fraction? Thank very much for this comment. Low case notification is children is mainly due to lack of accurate and reliable diagnostic tools, making diagnosis difficult to accomplish. Accessibility is an issue in this community, and the fact that diagnosis requires additional consideration of clinical evaluation by a professional could further hinder the process. ECEWS tried to hamper accessibility issues through providing transport vouchers for parents and absorbed the cost of chest x-rays for children. However, quality of diagnostic tools remains an important issue that we should continue to advocate for. We have added an explanation on this on lines 325 to 333 (page 8).
  • C004: P4L140-141. Data Analysis. Would you mind elaborating on the data extrapolation algorithm to predict registered TB cases based on historical data? The addition of a graph (scatter plot with regression line) would increase the readability of the manuscript. Thank you, we agree that adding further explanation on expected notifications would improve the manuscript. We have added an additional explanation between lines 204 to 211 (page 4). We have also added a chart on page 7 (Figure 2), accompanied by an explanation on 240 to 245 (page 5).
  • C005: Data acquisition / Project design. It is strongly recommended to provide some information about the randomization of (a) districts that received and did not receive the intervention (2) population groups that received different intervention strategies. Thank you for bringing up this point. The study was not randomized, rather he LGAs were selected in a purposive manner. The evaluation population was selected based on its location in the coastal areas of Nigeria and lack of access to TB diagnostic services. The control population was then matched by similarity in population size and access to health services (for which number of health facilities was used as a proxy). The control population was also selected in a different state to limit cross-over between both evaluation and control populations. We have elaborated on this on lines 80-88 (page 2).
  • C006: The authors provide a thoughtful discussion about implementing the active screening strategies, weighing both the benefits and restraints, especially regarding the availability of screening resources. A 2-fold increase in incidence by the least intensive approach, i.e., the community outreach, is an impressive number. It would be fascinating for the readers to discuss the direct implication of this finding in the Public Health organization of Nigeria and provide a financial outlook on how the state of Nigeria or the sponsors of this research are planning to proceed.   As this was a proof-of-concept intervention, the initial idea was to see if the approach would work in this setting. Given the subsequent success, ECEWS received a second year of funding from TB REACH to scale up the intervention to 15 LGAs across 5 states. We hope that disseminating the results of this intervention will encourage further investment on active case finding in rural Nigeria by the government and other sponsors, as well as encourage similar implementation in other high TB burden countries. ECEWS’ approach is also currently being considered for inclusion into Nigeria’s National Strategic plan for tuberculosis. We have added a few sentences to illustrate this at the end of the conclusion (line 401 to 405; page 9).

Thank you for your consideration of our manuscript.

Best,

Dr. Valerie Obot.
